# p53 Isoforms as Cancer Biomarkers and Therapeutic Targets

**DOI:** 10.3390/cancers14133145

**Published:** 2022-06-27

**Authors:** Liuqun Zhao, Suparna Sanyal

**Affiliations:** Department of Cell and Molecular Biology, Uppsala University, SE-75124 Uppsala, Sweden; liuqun.zhao@icm.uu.se

**Keywords:** p53 isoforms, cancer, biomarker, therapeutic target

## Abstract

**Simple Summary:**

The well-known tumor suppressor protein p53 plays important roles in tumor prevention through transcriptional regulation of its target genes. Reactivation of p53 activity has been a potent strategy for cancer treatment. Accumulating evidences indicate that p53 isoforms truncated/modified in the N- or C-terminus can modulate the p53 pathway in a p53-dependent or p53-independent manner. It is thus imperative to characterize the roles of the p53 isoforms in cancer development. This review illustrates how p53 isoforms participate in tumor development and/or suppression. It also summarizes the knowledge about the p53 isoforms as promising cancer biomarkers and therapeutic targets.

**Abstract:**

This review aims to summarize the implications of the major isoforms of the tumor suppressor protein p53 in aggressive cancer development. The current knowledge of p53 isoforms, their involvement in cell-signaling pathways, and their interactions with other cellular proteins or factors suggests the existence of an intricate molecular network that regulates their oncogenic function. Moreover, existing literature about the involvement of the p53 isoforms in various cancers leads to the proposition of therapeutic solutions by altering the cellular levels of the p53 isoforms. This review thus summarizes how the major p53 isoforms Δ40p53α/β/γ, Δ133p53α/β/γ, and Δ160p53α/β/γ might have clinical relevance in the diagnosis and effective treatments of cancer.

## 1. Introduction

p53 was discovered in 1979 as a 53 kD protein bound to simian virus (SV40) large T-antigen [1,2]. The p53 protein is a DNA sequence-specific transcription factor that suppresses tumor growth by regulating a large number of genes involved in cell cycle arrest, DNA repair, senescence, apoptosis, cell motility, adhesion, and migration [3,4,5]. p53 is the most frequently mutated gene in human cancers [6]. Loss or alteration of p53 functions for binding to p53 response elements located in its targeted genes leads to the development of cancer [7,8], thereby strengthening its significance in tumor suppression.

The p53 protein is a product of the TP53 gene that is highly conserved through evolution [9]. The human TP53 gene is located on the human chromosome 17p13.1 [10]. It is comprised of 11 exons, including 10 exons coding for the DNA sequence of the canonical or full-length p53 protein (FLp53, also termed p53α) and the first noncoding exon (exon 1) (Figure 1A) [11,12]. The TP53 gene expresses at least nine different mRNA transcripts encoding at least twelve different protein isoforms, namely p53α, p53β, p53γ, Δ40p53α, Δ40p53β, Δ40p53γ, Δ133p53α, Δ133p53β, Δ133p53γ, Δ160p53α, Δ160p53β, Δ160p53γ. These isoforms are the results of transcription initiation from different promotors, alternative splicing of the introns, and translation initiation at different codons [13,14].

It is interesting to note that although alternative splicing of p53 introns was first reported in the mid-1980s, the field of p53 isoforms did not emerge until quite recently [15,16]. In contrast, for proteins p63 and p73, both p53 homologues, the isoforms were identified almost immediately after their own discovery in the late 1990s [17,18,19]. However, the discovery of p63 and p73 isoforms accelerated the process of characterizing the first p53 isoform Δ40p53 [20]. Since then, studies on the p53 isoforms had been developing rapidly.

FLp53 consists of 393 amino acids and contains seven functional domains (Figure 1B) [14,21]. The two transactivation domains (TAD-1 and -2) are responsible for activating gene transcription [22]. The proline-rich domain (PRD) contributes to p53-mediated apoptosis [23]. The core DNA binding domain (DBD), as the name suggests, plays a key role in binding to specific DNA sequences [24]. The nuclear localization signaling region (NLS) is responsible for delivering p53 into the nucleus [25]. The oligomerization domain (OD) is involved in the formation of p53 tetramer [26]. Finally, the negative regulation domain (α) plays a crucial role in p53 post-translational modification that modulates the stability and activity of p53 [27,28].

The major p53 isoforms are truncated either in the N-terminus or in the C-terminus with some sequence alteration. The Δ40p53 isoform, without the first 40 amino acids, lacks the TAD-1 but still contains the TAD-2, which likely allows it to bind to both typical and atypical responsive elements [13,20]. The Δ133p53 and Δ160p53 isoforms without the first 133 and 160 amino acids, respectively, lack both transactivation domains TAD-1 and -2, and also a part of DBD supposedly modulating DNA binding property of p53 [13,21,22]. Other than these N-terminally truncated isoforms, p53 has two distinct C-terminally shortened variants (β and γ) [13,29]. In these isoforms, the OD is replaced by a stretch of 10 and 15 amino acids, with sequences DQTSFQKENC and MLLDLRWCYFLINSS respectively (Figure 1B) [29]. Given that the truncated p53 isoforms may have different structures from FL p53, it can be anticipated that they would display altered functions in relation to tumor suppression and can be even oncogenic [13,30,31].

Over the last decade, there has been a lot of evidence regarding the expression of p53 isoforms in various common cancers [13]. High p53β expression was positively associated with recurrence-free survival (RFS) and overall survival (OS) in patients with renal cell carcinoma [32]. Similarly, high levels of Δ133p53 expression have been correlated with better OS in patients with ovarian cancer [33,34]. In breast cancer, the most common type of female cancer, a low Δ40p53-to-FLp53 ratio is associated with increased disease-free survival (DFS) [35,36]. On the other hand, Δ40p53α overexpression exerts tumor suppressor activity by increasing the expression of p53 target genes in the most common type of primary liver cancer hepatocellular carcinoma [37]. Moreover, amyloid aggregation of Δ40p53 has been reported in the endometrial cancer cells [38]. The core domain of p53 also demonstrates aggregation propensity, particularly with the influence of various RNAs (mRNA and rRNA), which may have implications in cancer [39,40]. Treatment with the histone deacetylase inhibitor valproic acid resulted in downregulation of p53β/γ and upregulation of p53α in acute myeloid leukemia, suggesting that p53 isoforms have the potential to predict therapy response [41]. p53 isoforms, especially the N-terminally truncated ones Δ40p53, Δ133p53, and Δ160p53, play an important role in the development of cancers [13]. Altogether, p53 isoforms play critical roles in tumorigenesis and might be potential biomarkers and therapeutic targets.

## 2. Regulatory Mechanisms of p53 Isoform Expression

Current literature shows that the expression of the p53 isoforms is regulated by different mechanisms at different levels in the cells. The expression of the p53 isoforms is regulated at the transcriptional, post-transcriptional, translational, and post-translational levels. Moreover, different degradation mechanisms exist for different p53 isoforms, which also modulate their relative levels.

A major regulation occurs in the level of transcription by alternative promoter usage and alternative splicing [14], as elaborated below. TP53 gene has two different promoters, P1 and P2 (Figure 2A). FL p53 and Δ40p53 are encoded from the constitutive promoter P1 [42], whereas Δ133p53 and Δ160p53 mRNAs are transcribed from the promoter P2 [43,44]. 17β estradiol regulates p53 isoform expression through activating c-Myc that induces the P1 promoter of the TP53 gene in breast cancer cells [45]. Alternatively, p53 has been shown to increase the transcription of Δ133p53 by binding to the p53 response elements in the internal P2 promoter [44,46]. In addition, several p63/p73 isoforms such as p63β, ΔNp63α, ΔNp63β, and ΔNp73γ activate Δ133p53 expression by transactivating the promoter P2 [47].

Alternative splicing of the p53 mRNA plays a major role in the expression of the p53 isoforms. Differential splicing of intron 2 (i2) generates two different transcripts that encode the isoforms FLp53 and Δ40p533, respectively (Figure 1) [48]. A G-quadruplex structure in intron 3 (i3) regulates the splicing of i2 to generate the mRNAs for either FLp53 or the Δ40p53 isoform (Figure 2A) [49]. Alternatively, the mRNAs for the p53 α/β/γ are generated by alternative splicing of intron 9 (i9) (Figure 1A) under regulation of the serine/arginine-rich (SR) splicing factors (SRSFs) such as SRSF1 and SRSF3 (Figure 2A). Downregulation of SRSF1 stimulates transcription of the p53β and p53γ isoforms [50]. In contrast, SRSF3 downregulation only increases p53β [51]. Thus, alternative splicing mechanisms can be used to induce p53 isoform expression.

p53 isoform expression can also be regulated at translational and post-translational levels. The Δ40p53 isoform can be generated from the p53 transcript by alternative initiation of translation at the ATG40 codon. In that case, the stop codons retained in i2 of the p53 transcript can result in termination of translation followed by re-initiation at the AUG40 codon, thereby expressing the Δ40p53 isoform (Figure 1A and Figure 2A) [48]. FLp53 (p53α) and Δ40p53 expression are regulated by two distinct internal ribosome entry sites (IRESs) (Figure 2A) [52,53]. Multiple IRES-interacting trans-acting factors (ITAFs) have been found to regulate IRES-mediated translation of p53 mRNAs. These include polypyrimidine-tract-binding protein (PTB), PTB-associated splicing factor (PSF), Annexin A2, death-associated protein 5 (DAP5), translational control protein 80, and RNA helicase A [54,55,56,57,58,59]. Therefore, factors mediating expression of ITAFs also play a role in p53 isoform expression. The Δ133p53 and Δ160p53 isoforms are expressed from the same mRNA transcript. However, the Δ160p53 isoform is produced by alternative initiation of translation at ATG160 from the Δ133p53 transcript [43].

The stability of different p53 isoforms in the cells are also regulated by p53 degradation pathways. Under normal cellular conditions, p53 is regulated via a negative feedback loop involving its target gene E3 ubiquitin ligase mouse double minute 2 (MDM2). MDM2 binding to p53 TADs inhibits p53 activity by preventing it from regulating the target genes [60,61]. Alternatively, p53 can induce the expression of MDM2, thus forming a negative feedback loop [62]. Binding of MDM2 to p53 promotes its rapid degradation through ubiquitination using 26S proteasome pathway (Figure 2B, middle panel) [63,64]. The N- terminally truncated p53 isoforms lack the TADs partially or fully and thus may not be fully susceptible to MDM2 based ubiquitination. Earlier reports suggest that the Δ40p53 isoform can escape from MDM2-induced protein degradation [65]. However, another study reports that by retaining the DBD, and the C-terminal domain, the N-terminally truncated p53 isoforms can still bind to MDM2, which can be enough for p53 ubiquitination and proteasomal degradation [66]. It has also been demonstrated that Δ40p53 can protect p53 from MDM2-mediated degradation by interfering with p53-MDM2 binding (Figure 2B, middle panel) [65,67].

The p53β/γ isoforms tend not to be degraded by MDM2-mediated ubiquitination, although they contain TADs. The p53β/γ mRNAs are susceptible to nonsense-mediated decay (NMD) that can degrade mRNAs with premature translation termination codons (PTCs) (Figure 2B, left panel) resulting from alternative splicing of p53 i9 (Figure 2A) [68]. Thus, the expression of p53β and p53γ can be regulated at the post-transcriptional level.

A recent study suggests that the stability of the Δ133p53 isoform is not affected by a proteasome inhibitor, MG132 [69]. Δ133p53 can be degraded by the digestive organ expansion factor (Def) protein in a proteasome-independent degradation pathway [70]. Δ133p53α can also be degraded by autophagy during replicative senescence (Figure 2B, right panel) [71]. Downregulation of the chaperone-associated ubiquitin ligase STUB1 causes Δ133p53α autophagic degradation [71]. To the best of our knowledge, there have been no reports yet of the regulation of Δ160p53 degradation. Δ40p53 can also be generated post-translationally via a 20S proteasome-mediated cleavage of FLp53 [72]. Taken together, at least four levels of regulation govern the expression of p53 isoforms: transcriptional, post-transcriptional, translational, and post-translational.

## 3. Influence of the p53 Isoform Network on Biological Processes

The p53 network takes a central position in regulation of different cellular activities, but it is also regulated by external or internal factors. The p53 pathway is stimulated by various cellular stress factors such as DNA damage, cell contact, oncogene expression, oxidative stress, and endoplasmic reticulum (ER) stress (Figure 3). These in turn lead to multiple cellular effects including cell cycle arrest, senescence, cellular apoptosis, and DNA damage repair, resulting in tumor suppression [21]. p53, also called the ‘guardian of the genome’, acts as a transcription factor, regulating expressions over 3600 target genes [73,74]. Thus, the p53 pathway can play a critical role in tumor suppression by activating or repressing target genes [75].

Many studies have shown that p53 isoforms influence diverse biological activities of p53 by regulating p53 target genes. p53 isoforms can also interact with other specific proteins or factors to modulate the p53 pathway. For example, Δ133p53β can regulate apoptosis by inhibiting the activity of an anti-apoptotic protein, RhoB [76]. Netrin-1, an anti-apoptotic protein, can be positively regulated by Δ40p53 [77]. Similarly, overexpression of Δ133p53 has been shown to improve DNA double-strand break (DSB) repair by upregulating E2F1, a transcription factor that activates the DNA DSB repair factor RAD51 [78,79]. These studies demonstrated that p53 isoforms can mediate their network without relying on p53. Furthermore, several signaling pathways interact with the p53 pathway in a p53-isoforms-dependent manner (Figure 4) [75,80].

### 3.1. p53 Isoforms Regulating Cell Cycle Arrest and Senescence

Cellular senescence is usually used to describe an irreversible cell cycle arrest [81]. 14-3-3σ is a member of 14-3-3 family proteins that interact with p53 and affect its activity [82]. It has been shown that Δ40p53 can form a homotetramer complex and bind to the promoter of 14-3-3σ, activating the expression of 14-3-3σ and causing cell cycle arrest in response to ER stress [83]. Thus, the interaction between Δ40p53 and 14-3-3σ can play a role in cell cycle arrest. p53 can also regulate cell cycle arrest by activating its target gene p21 [84,85]. In addition, p53-induced production of microRNA-34a (miR-34a) leads to gene expression involved in cell cycle arrest and senescence in oncogenic cells [86,87]. Both p21 and miR-34a can act as the pro-senescence factors [30]. Ota et al. showed that Δ40p53 expression induced tumor cell senescence by upregulating p21 in hepatocellular carcinoma cells (Figure 3) [37]. In addition, overexpression of Δ133p53 induced cell cycle arrest and cellular senescence when exposed to oxidative stress by H_2_O_2_ in mouse embryonic fibroblasts [88]. Besides that, Δ133p53 expression inhibits senescence by repressing the expression of p21 and miR-34a (Figure 3) [69,89,90]. This is because Δ133p53 can prevent p53α from binding to the promoters of p21 and miR-34a [89]. Furthermore, p53β induces cellular senescence through forming complexes with p53α and upregulating p21 expression [69,91]. Thus, p53 isoforms modulate cell cycle arrest and senescence via inducing or suppressing p53 target genes.

### 3.2. p53 Isoforms—The Role in Apoptosis

The Bcl-2 protein family, which includes Bcl-2, Bcl-xL, Bax, NOXA, and PUMA, is important for cell apoptosis [92]. p53 is known to induce apoptosis by inhibiting the expression of the anti-apoptotic proteins such as Bcl-2 and Bcl-xL and activating pro-apoptotic proteins such as Bax, NOXA, and PUMA [92,93,94]. In addition, p53-induced protein with a death domain (PIDD) can promote p53-mediated apoptosis [95]. Several p53 isoforms have been linked to the regulation of apoptosis. For instance, p53β can form a complex with p53α and boost p53 transcriptional activity on the Bax promoter, thereby enhancing apoptosis (Figure 3) [91]. Alternatively, Δ133p53/Δ113p53 has been shown to inhibit p53-mediated apoptosis through activating Bcl-xL in zebrafish (Figure 3) [96]. In response to low or moderate stress, Δ133p53 and Δ123p53 (Δ133p53 mouse ortholog) can form a complex with p53α and switch the p53 binding sites within the Bcl-2 promoter from repressive to activating p53 response elements, thereby resulting in Bcl-2 induction and anti-apoptosis (Figure 3) [97]. These studies suggest that Δ133p53 prefers to be an antagonist of p53-mediated apoptosis. In contrast, when the expression of Δ40p53 is lower or equal to that of p53α, it induces the expression of Bax and PIDD, resulting in apoptosis (Figure 3) [98,99]. Together, these results suggest that the p53 isoforms influence apoptosis primarily through interactions with p53α.

### 3.3. p53 Isoforms and DNA DSB Repair

DNA DSBs are one of the most harmful types of DNA damage [100]. Correct DNA DSB repair is important for prohibiting tumorigenesis [100]. It is known that p53 is activated in response to DNA damage [101], which, in turn, inhibits DNA DSB repair [102] by downregulating DNA DSB repair genes including RAD51, RAD52, LIG4, WRN, and XRCC4 [103]. Interestingly, the p53 isoform Δ133p53 plays a significant role in promoting DNA DSB repair [104]. Δ133p53/Δ113p53 (zebrafish ortholog of Δ133p53) can switch p53 from repression to activation DNA DSB repair via upregulating the expression of RAD51, RAD52, and LIG4 (Figure 3) [103,105,106]. In addition, upon γ-irradiation, Δ133p53 can form a complex with p73 and can stimulate the expression of genes RAD51, RAD52, and LIG4 with higher efficiency [107]. Thus, with a prominent role in DNA DSB repair, Δ133p53 can be strongly associated with tumorigenesis.

### 3.4. p53 Isoforms and NF-κB Signaling

Nuclear factor-κB (NF-κB) has both positive and negative effects on tumor progression [108]. The interaction of two critical signaling pathways, NF-κB and p53, can regulate several important cellular processes involved in tumorigenesis, including senescence, apoptosis, inflammation, and DNA damage repair [109]. Using a luciferase reporter assay, it has been shown that Δ133p53 can induce NF-κB activity upon infection by *Helicobacter pylori*, a strong risk factor for gastric cancer [110]. This, in turn, induces the expression of the NF-κB target genes including anti-apoptotic protein Bcl-2 and pro-inflammatory cytokines interleukin 6 (IL-6) and IL-8 (Figure 4) [110]. However, NF-κB is blocked in p53-null cells [110,111]. Thus, Δ133p53 activates NF-κB activity in a p53-dependent manner, which would inhibit apoptosis and promote inflammation during *H. pylori* infection (Figure 4). Another study reported that Δ133p53 mRNA level was downregulated upon treatment with NF-κB inhibitor pyrrolidine dithiocarbamate in MKN45 gastric cancer cells [112]. Thus, Δ133p53 and NF-κB expression levels are positively correlated, and downregulation of Δ133p53 may disrupt NF-κB signaling, which is significant for tumor development.

### 3.5. p53 Isoforms, JAK- STAT and RhoA-ROCK Signaling

Janus kinase (JAK)-signal transducer and activator of transcription (STAT), commonly called JAK-STAT signaling, is closely associated with cancer progression [113]. The JAK kinases activate the STAT family proteins through phosphorylation of tyrosine and serine residues [114,115]. STAT3, a member of the STAT family, is a transcription factor that can bind to various gene promoters [116]. These genes can be involved in various cellular processes such as anti-apoptosis and cell cycle progression (Figure 4) [114,116]. Of interest, STAT3 is necessary for the activity of NF-κB [117]. Both STAT3 and NF-κB induce the expression of the genes involved in anti-apoptosis, such as Bcl-2 and Bcl-xL, and cell cycle progression, such as c-Myc [116,118]. Thus, the activation of JAK-STAT and NF-κB signaling pathways contributes to cancer progression. Furthermore, IL-6 is considered as an activator of JAK-STAT signaling (Figure 4) [113]. As mentioned above, Δ133p53 induces the expression of NF-κB target genes including IL-6 [110]. Thus, Δ133p53 may activate the JAK-STAT signaling pathway by interacting with NF-κB signaling.

RhoA-ROCK signaling pathway consists of a small GTPase protein RhoA and its effector Rho-kinase (ROCK) [119]. RhoA cyclically exchanges between an active GTP-bound state and an inactive GDP-bound state (Figure 4) [120]. Interestingly, RhoA can drive STAT3 activation [121], suggesting an interplay between the RhoA with STAT. Furthermore, JAK activates ROCK; in turn, ROCK induces STAT3 phosphorylation (Figure 4) [122,123]. A study has shown that the increased expression of Δ133p53 activated the RhoA-ROCK signaling pathway probably by activating IL-6 and, further, by inducing the cooperation between JAK-STAT and RhoA-ROCK pathways [123]. Altogether, Δ133p53 expression can regulate the coordinated action of the NF-κB, JAK-STAT, and RhoA-ROCK signaling pathways (Figure 4).

### 3.6. p53 Isoforms and IFN Signaling

Interferons (IFNs) are pleiotropic cytokines that have important functions/properties including antiviral, antiproliferative response, immunoregulation, and antitumor [124,125]. There are three types of IFNs in humans: type I (IFN-α/β), type II (IFN-γ), and type III (IFN-λs) [124,126,127]. Several studies have shown that p53 isoforms are involved in IFN signaling [128,129,130,131]. Both p53β and p53γ isoforms were shown to be involved in the induced production of IFN-α and IFN-β, which have a critical role in the antiviral response [131,132]. In turn, IFN-β can induce both FLp53 (p53α) and p53β/γ isoforms in human peripheral blood mononuclear cells, potentially altering the expression of p53 target genes involved in cell cycle arrest or apoptosis [133]. Type II IFN IFN-γ is associated with antiviral, antiproliferative, immunomodulatory, and antitumor responses [124,134,135]. In estrogen-receptor-positive breast cancer with mutant p53, Δ133p53 can activate the expression of IFN-γ signaling genes [129]. Additionally, overexpression of Δ133p53β increased the expression of genes involved in the IFN-γ signaling pathway in prostate cancer [128]. Thus, Δ133p53 isoforms could activate IFN-γ signaling in different cancer types. Moreover, IFN-α/β/γ can induce the JAK-STAT signaling pathway that plays an important role in tumor progression (Figure 4) [126,130,136]. Collectively, these studies suggest that p53 isoforms that interact with the IFN signaling pathway have an important influence on antiviral immune responses and the risk of developing cancer.

Except for wild-type p53, Δ133p53 is the most well studied p53 isoform. In comparison, little is known about Δ160p53 and p53γ activity in modulating the p53 signaling pathway. As a result, much research remains to be done in the field of the p53 isoform network. Collectively, the biological activities of p53 signaling and several other signaling pathways can be affected by p53 isoforms.

## 4. p53 Isoforms as Promising Cancer Biomarkers

Many clinical studies have attempted to evaluate the clinical relevance of the p53 isoforms in various types of human cancers, both at the mRNA and the protein levels [13]. Endogenous p53 isoforms are generally found to be overexpressed in tumors when compared to nontumor cell lines, making p53 isoforms promising cancer biomarkers. In human cells, p53 isoforms express tissue specifically at both the mRNA and protein levels [21,91]. It is therefore important to quantify the expression levels of p53 isoforms in cells and tissues to be able to use those as cancer biomarkers.

To detect the p53 isoforms, several p53 isoform-specific antibodies that recognize different epitopes have been developed. However, these antibiotics lack specificity due to the similar epitopes in other p53 isoforms [78]. Due to unavailability of the antibodies specific to each p53 isoform, the expression of different p53 isoforms is mostly quantified at the mRNA level using RNA-seq [137,138]. The abundance of TP53 transcripts is quantified in many clinical studies using short-read RNA sequencing, which cannot distinguish 5′ variants of the α, β, and γ transcripts [139,140]. Recently, several studies have developed new approaches to quantitate distinct transcripts, such as RNAscope and multiplex probe-based long amplicon droplet digital PCR [128,140,141]. However, the correlation between the amount of the mRNAs and the abundance of their corresponding proteins can be weak, even for general proteins [142,143]. Furthermore, RNA-seq analysis has a disadvantage in detecting less-abundant p53 isoform transcripts [138]. To detect each C-terminal p53 isoform, a new technique combining molecularly imprinted polymers (MIPs) and LC-MS/MS-based targeted proteomics has recently been developed [144]. Thus, the development of novel and highly specific anti-p53 isoform approaches is very desirable.

Deregulation of p53 isoforms can either promote or inhibit tumor progression in a variety of cancers. Following the literature, different p53 isoform expression profiles have been correlated with different cancer types (Table 1 and Table 2). So far, only a few studies have shown that p53 isoforms expression can slow down tumor progression (Table 1). For example, high p53β expression is associated with a good prognosis in clear-cell renal cell carcinoma [32]. Similarly, high Δ40p53 expression has been inferred to be a prognostic marker for recurrence-free survival (RFS) in mucinous ovarian cancer [145]. High Δ40p53α expression has been reported to inhibit tumor cell growth and, at the same time, increase cellular senescence in hepatocellular carcinoma [37]. Moreover, high ∆133p53 mRNA expression levels are found to be a predictor of improved overall survival (OS) in patients diagnosed with high-grade serous ovarian carcinoma [34]. The p53 isoform expression profile in myeloma has recently been identified [146]. In these patients with high-risk cytogenetics, high expression of the short p53 isoforms Δ133p53 and Δ160p53 has been reported to better OS [146]. Furthermore, in brain cancer patients, ∆133p53 expression helped to reduce the side effects of radiation, such as astrocyte senescence and astrocyte-mediated neuro-inflammation [79]. These studies, thus demonstrate that high levels of p53 isoform expression can be used as a survival-associated biomarker for several types of cancer.

The p53 isoforms are often reported to promote tumor progression in a variety of cancers (Table 1). The Δ40p53 isoform has been identified as the major component of p53 amyloid aggregates in endometrial carcinoma cancer. It plays an important role in modifying p53 aggregation properties, which is associated with tumor progression [38]. The increased expression of Δ133p53β has been predicted to have poorer outcomes in glioblastoma, melanoma, breast, and prostate cancer patients [128,141,147,148]. Additionally, elevated Δ133p53 mRNA levels were thought to be a potential marker for lung and colorectal cancer malignancy [123,149]. A recent study also revealed that overexpression of Δ160p53 can stimulate the proliferation and migration of melanoma cells [150]. Besides, the N-terminally shortened p53 isoforms, the C-terminal truncated variants p53β and -γ can also aid in cancer prognosis. In uterine serous carcinoma, the high expression of p53γ has been associated with reduced progression-free survival (PFS) [151]. Moreover, high levels of p53β and -γ expression had a negative effect on the survival of myeloma patients [146]. These findings suggest that high p53 isoform expression could be a potential biomarker for cancer survival outcome.

Other than high or low expressions, proper Δ40p53-to-p53α ratio can be important for suppressing cancer [67]. In breast cancer, a high Δ40p53-to-p53α ratio is associated with poorer outcomes (Table 2) [152]. Patients with cholangiocarcinoma who have a high ratio of Δ133p53-to-p53α have a poor prognosis (Table 2) [153]. Furthermore, a high ∆133p53-to-p53β ratio increases cancer aggressiveness in colorectal cancer (Table 2) [69]. In acute myeloid leukemia, a high degree of p53β/γ-to-p53α predicts a better prognosis (Table 2) [154]. These studies suggest that the ratio of p53 isoforms can also be a potential biomarker in cancer and can be involved in clinical cancer therapy.

Mutations in the TP53 gene are seen to occur in ~50–60% of human cancers [7]. The majority of these missense mutations in the DBD cause p53 to lose its tumor-suppressive function and acquire novel oncogenic functions [7,155,156]. There are a few reports which suggest that p53 isoforms are beneficial for inhibiting the oncogenic activities of the p53 mutants. For example, Δ133p53 isoform can interact with the mutant FLp53 and have a beneficial effect in advanced serous carcinomas [157]. In addition, Δ133p53 RNA can upregulate the expression of IFN-γ signaling genes in mutant TP53 oestrogen receptor positive breast cancer, which is associated with a better patient outcome [129]. Furthermore, when compared to patients with wild-type p53 breast cancer, patients with mutant p53 breast cancer who expressed the p53γ isoform had a lower risk of cancer recurrence and a better OS [158]. In contrast, high levels of ∆160p53 expression contribute to the proliferation of cancer cells with p53 mutations [159]. Based on current evidence, it can be stated that the p53 isoforms can be prognostic biomarkers in human cancer.

## 5. p53 Isoforms as Potential Therapeutic Target in Cancer

The therapeutic values of p53 isoforms are yet to be investigated. The p53 isoforms have good potential as therapeutic targets in cancer due to their various regulatory expression mechanisms, complex interaction network, and different expression levels in tumor versus nontumor cells. Because relative expressions of the p53 isoforms are relevant in various cancers, the factors affecting the ratio of the p53 isoforms can also be therapeutic targets. For example, ITAFs, which regulate the IRESs responsible for the relative expression of FLp53 and ∆40p53 [52], can be targets for cancer treatment [160]. However, more research on ITAFs will be required to identify the specific targets for the first and the second IRES in order to use those for cancer therapy.

The aberrant alternative splicing of p53 mRNA has shown to be a hallmark of cancer [161,162]. Recently, three major therapeutic approaches, namely, small molecule inhibitors, splice-switching antisense oligonucleotides (SSOs), and clustered regularly interspaced short palindromic repeats (CRISPR)-Cas system, have been developed to target the splicing factors, providing a new strategy for cancer treatment [161,163]. Small molecules that affect the phosphorylation of serine/arginine-rich (SR) splicing factors (SRSFs) seem to have the potential to modify isoform production and provide therapeutic benefits [164,165]. Caffeine has been shown as a small molecule that downregulates the expression of the splicing factor SRSF3, altering splicing pattern of the gene TP53 and also other SRSF3 target genes [51,166]. An appropriate dose of caffeine decreases p53α expression while increasing p53β expression, which improves cellular senescence. Thus, caffeine can be used to modulate the p53β-to-p53α ratio, a ratio that is important for tumor treatment [51,69,166]. The SR proteins, which regulate alternative splicing of the p53 mRNA, are regulated by SR-protein-specific kinases such as Cdc2-like kinases (Clks), which phosphorylate SR proteins [167,168,169]. Thus, small molecules which can inhibit Clks may contribute to cancer treatment. For example, the Clks inhibitor TG003 can induce the expression of p53β and -γ in the MCF7 breast cancer cell line, which promotes cell apoptosis [50]. Other Clks inhibitors, such as KH-CB19 and GPS167, have also been shown to inhibit the phosphorylation of splicing factors [170,171] and thus can have therapeutic value. However, targeting Clks with small molecules is still facing clinical implementation challenges [163].

SSOs are short (15 to 30 nucleotide), chemically-modified RNA molecules which compete with splicing factors for binding to pre-mRNAs [161,163,172]. Though it is not yet approved for clinical trials, research is ongoing in cancer model systems to implement this strategy in cancer treatment [163]. In addition, the CRISPR-Cas gene-editing system has also been developed for alternative splicing of target mRNAs [163,173]. There is no doubt that splicing modulation is an important strategy for cancer treatment. However, more research is needed in order to implement that as an effective and safe therapeutic tool.

The mechanisms of p53 isoform degradation and aggregation can play an important role in regulating the expression of p53 isoforms. Gudikote et al. showed that upregulating p53β and -γ by inhibition of NMD can restore p53 activity in p53-deficient tumors caused by MDM2 overexpression or by inactivating mutations downstream of exon 9 [68]. There are mainly two reasons to consider NMD inhibition as a therapeutic strategy for p53 reactivation.

First, while p53β and -γ are significantly less susceptible to MDM2-mediated degradation, they are highly susceptible to NMD. Second, they intrinsically promote p53 transcriptional activity [68,91,174]. Furthermore, a peptide covering residues 107–129 of wild-type p53 could inhibit the aggregation of destabilized Δ133p53β and restore it to the wild-type p53 conformation, simultaneously restoring p53’s tumor suppressor activity [175]. Thus, the available p53-reactivation strategies could provide effective cancer therapies, highlighting the attractiveness of p53 isoforms as a therapeutic target. In terms of autophagic degradation of ∆133p53α, knockdown of autophagy-related proteins and a regulator protein STUB1, respectively, can decrease and increase the degradation process [71]. This strategy opens up a new avenue for regulating the abundance of ∆133p53α via autophagic degradation. Overall, modulating the stability of p53 isoforms may be a useful tool for cancer treatment.

Recently, Sun et al. reported that ∆40p53 expression is positively correlated with netrin-1, a cancer biomarker and therapeutic target protein that can inhibit apoptosis in several aggressive cancers [77,176]. Netrin-1 inhibition has been shown to be a highly promising strategy in human cancers with high levels of ∆40p53 expression [77]. Similarly, patients whose cancers have high Δ133p53 expression may benefit from JAK-STAT signaling inhibitors [123,177,178]. Thus, indirect modulation of p53 isoform expression by specific factors has the potential to be used as a therapeutic strategy in tumors expressing p53 isoforms.

## 6. Conclusions

The tumor suppressor protein p53 has been well studied in terms of its role in cancer development and progression. The TP53 gene does not only express just one protein p53; at least twelve different p53 isoforms are produced by alternative promoter usage, alternative splicing, and alternative translation initiation events. Current studies have revealed that p53 isoform expression is regulated at several levels, including transcription, post-transcription, translation, and post-translation. The distinct expression profiles of p53 isoforms in tumor and normal cells, as well as the deregulation of p53 isoforms as an emerging contributor to cancer development, make p53 isoforms biomarkers in cancer. Alternative-splicing-mediated transcription and IRES-mediated translation are important tools for regulating the expression of p53 isoforms, which would likely provide attractive strategies for cancer therapeutics. Furthermore, the intricate p53 isoform network modulates many biological processes, cooperation with other proteins, etc. Thus, a thorough understanding of the p53-isoform-mediated biological processes can open up new avenues for cancer therapeutics.

## Figures and Tables

**Figure 1 cancers-14-03145-f001:**
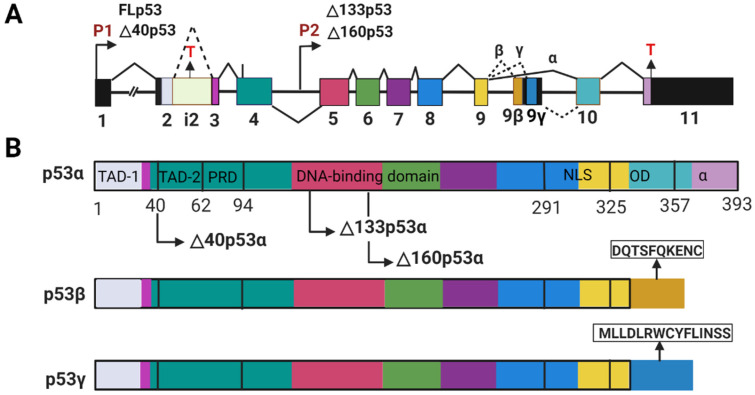
Constitution of the human TP53 gene and p53 isoforms. (**A**) Schematic representation of different elements of the TP53 gene. The TP53 gene is composed of 11 exons marked with numbers 1–11. It has two promoter regions P1 and P2, which produce transcripts of different length for expression of different p53 isoforms. The RNA transcripts for FL53 and its isoforms are also generated by alternative splicing of the introns (i2 and i9), and alternative initiation of translation. The stop codons (T) present in i2 and exon 11 are indicated with T. (**B**) Schematic representation of the p53 isoforms. p53α (FLp53) has seven structural components: two transactivation domains (TAD-1 and -2), a proline-rich domain (PRD), a DNA binding domain (DBD), a nuclear localization signaling region (NLS), oligomerization domain (OD), and a negative regulation domain (α). Δ40p53, Δ133p53, and Δ160p53 isoforms have a deletion of 40, 133, and 160 amino acids from the N-terminus, respectively. Their start codons are indicated. The β and γ isoforms lack the OD and the α domains in the C-terminus, and instead have extensions DQTSFQKENC and MLLDLRWCYFLINSS, respectively.

**Figure 2 cancers-14-03145-f002:**
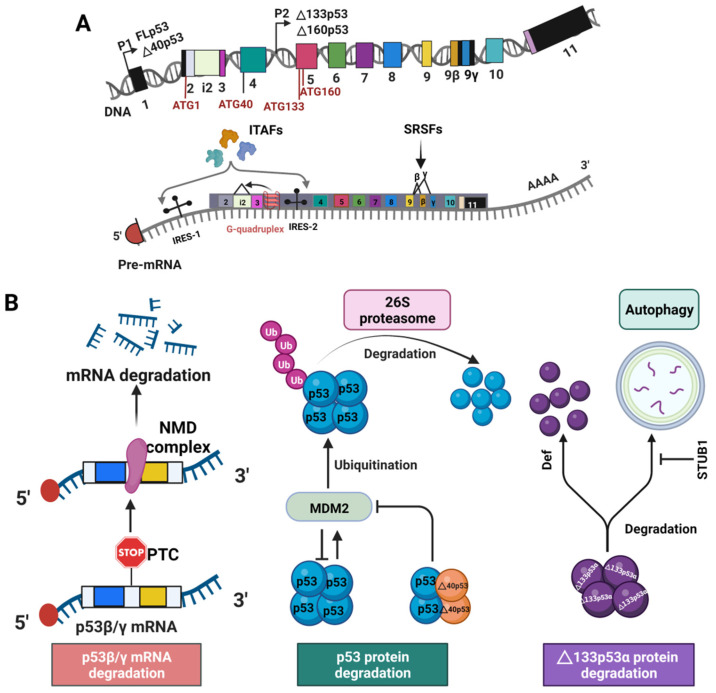
Expression of the p53 isoforms is regulated by different mechanisms. (**A**) Regulation at the levels of transcription and translation. At the transcriptional level, alternative promoter usage (P1 and P2) lead to the production of mRNAs for different p53 isoforms. Moreover, alternative splicing (Λ) of the introns i2 and i9, guided by G-quadruplex structures and serine/arginine-rich (SR) splicing factors (SRSFs), respectively, govern expression of the p53 N- and C- terminal variant isoforms. In addition, two distinct internal ribosome entry sites (IRESs) under action of the IRES-interacting trans-acting factors (ITAFs) can modulate the expression of p53 and Δ40p53 at the level of translation. (**B**) The degradation mechanisms of p53 and its isoforms. Nonsense-mediated decay (NMD) degrades the p53β and p53γ mRNAs post-transcriptionally. The MDM2-mediated ubiquitination-26S proteasome pathway degrades wild-type p53. Δ40p53 forms a heterotetramer with p53, which prohibits MDM2-mediated degradation. Proteasome-independent degradation pathways, such as the digestive organ expansion factor (Def) protein-mediated pathway and autophagy, can degrade Δ133p53α. STUB1, a chaperone-associated ubiquitin ligase, can protect Δ133p53α from autophagic degradation.

**Figure 3 cancers-14-03145-f003:**
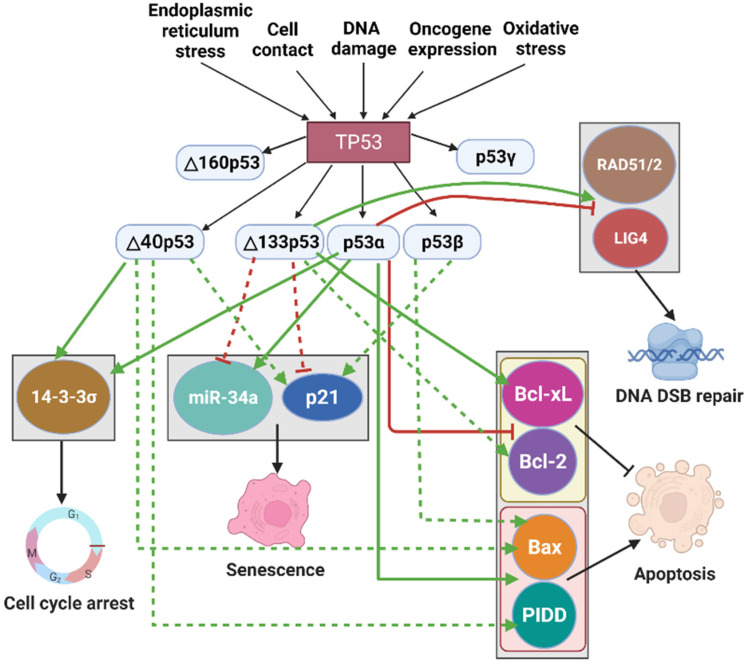
p53 isoforms modulate the p53-mediated cellular activities. Different cellular stresses such as ER stress, cell contact, DNA damage, oncogene expression, and oxidative stress activate the p53 signaling pathway. The TP53 gene produces multiple isoforms: p53α, p53β, p53γ, Δ40p53α/β/γ, Δ133p53α/β/γ, and Δ160p53α/β/γ. Isoforms Δ40p53, Δ133p53, and Δ160p53 have been reported to affect biological activities such as cell cycle arrest, senescence, apoptosis, and DNA DAB repair by inducing (green line) or repressing (red line) p53 target gene expression. Furthermore, p53 isoforms can regulate the expression of p53 target genes either directly (solid red and green line) or indirectly (dash red and green line). 14-3-3σ is a target gene of p53, and it induces cell cycle arrest. p53 can induce senescence by transactivating its target genes p21 and miR-34a. Activating p53 target genes such as Bcl-2 and Bcl-xL inhibits apoptosis, while p53 can induce apoptosis by activating pro-apoptotic proteins such as Bax and PIDD. Activating several p53 target genes such as RAD51, RAD52, and LIG4 contributes to DNA DSB repair.

**Figure 4 cancers-14-03145-f004:**
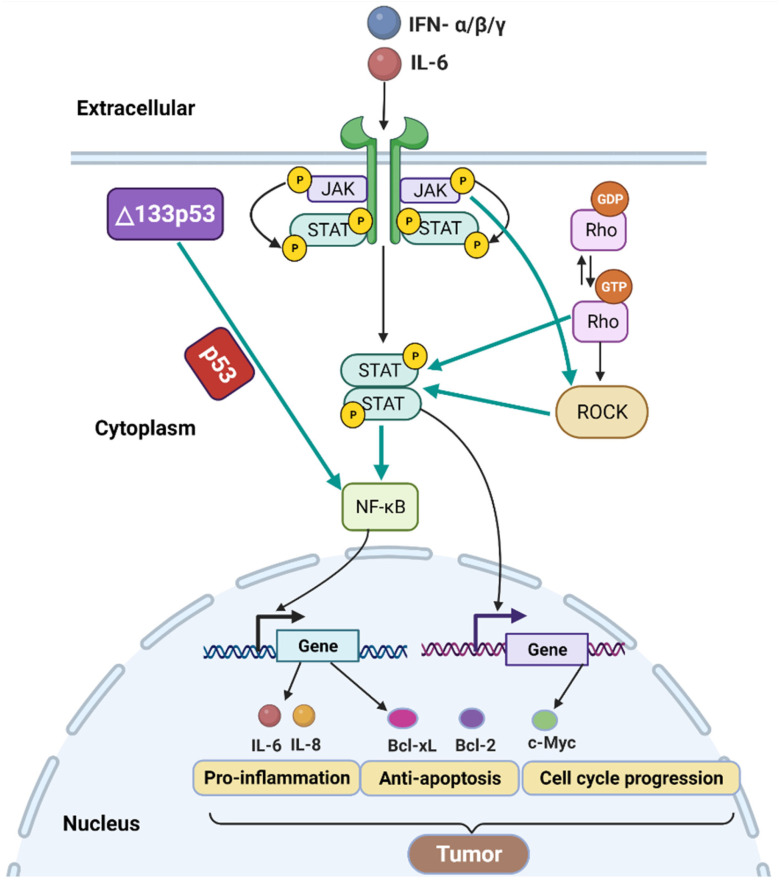
Mechanisms of Δ133p53-induced tumor development via several signaling pathways, including NF-κB, JAK-STAT, RhoA-ROCK, and IFN. In cancer cells, Δ133p53 expression induces the expression of NF-κB target genes, including IL-6, a key factor for activating phosphorylation of STAT. In addition, type I and II IFNs IFN-α, IFN-β, and IFN-γ can also activate the JAK-STAT signaling. STAT can stimulate the transcription of its target genes including Bcl-xL, Bcl-2, and c-Myc. STAT can also activate NF-κB. The activated NF-κB can also induce the transcription of various target genes, such as IL-6, IL-8, Bcl-xL, Bcl-2, and c-Myc. The RhoA-ROCK signaling pathway can interact with the JAK-STAT signaling pathway, resulting in increased phosphorylation of STAT. These pathways regulate genes involved in several processes, such as pro-inflammation, anti-apoptosis, and cell cycle progression, all of which aid in tumor development.

**Table 1 cancers-14-03145-t001:** The effects of increased expression of p53 isoforms on human cancer progression.

Tumor Types	Name of Proteins	Molecule Detected	Tumor Progression *	Reference
High-grade serous ovarian carcinoma	∆133p53	mRNA	↓	[34]
Mucinous ovarian cancer	∆40p53	mRNA and protein	↓	[145]
Hepatocellular carcinoma	∆40p53	protein	↓	[37]
Renal cell carcinoma	p53β	mRNA and protein	↓	[32]
Myeloma	∆133p53/∆160p53	mRNA and protein	↓	[146]
	p53β/γ	mRNA and protein	↑	[146]
Breast cancer	∆133p53β	mRNA and protein	↑	[147]
Melanoma	∆160p53	mRNA and protein	↑	[150]
	∆133p53β	mRNA and protein	↑	[148]
Colorectal cancer	∆133p53	mRNA	↑	[123]
Prostate cancer	∆133p53β	mRNA	↑	[128]
Glioblastoma	∆133p53β	mRNA	↑	[141]
Lung cancer	∆133p53	mRNA	↑	[149]
Endometrial carcinoma	∆40p53	mRNA and protein	↑	[38]
Uterine serous carcinoma	p53γ	mRNA	↑	[151]

* ↑ to accelerate tumor progression, ↓ to slow down tumor progression.

**Table 2 cancers-14-03145-t002:** The effects of high ratio of p53 isoforms to p53α/β on human cancer progression.

Tumor Types	NAME of PROTEINS	Molecule Detected	Tumor Progression *	Reference
Breast cancer	∆40p53:p53α	mRNA	↑	[152]
Colorectal cancer	∆133p53:p53β	mRNA	↑	[69]
Cholangiocarcinoma	∆133p53:p53α	mRNA	↑	[153]
Acute myeloid leukemia	p53β/γ:p53α	protein	↓	[154]

* ↑ to accelerate tumor progression, ↓ to slow down tumor progression.

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
