# Peer review of "p53 Isoforms as Cancer Biomarkers and Therapeutic Targets"

_cancers, 2022, doi:10.3390/cancers14133145_

Round 1

Reviewer 1 Report

The authors have made a considerable work of rewriting and addressed all my comments. The updated version of their manuscript is comprehensive and well written. I have no further concern.

Reviewer 2 Report

The authors have addressed all the raised points. No further comments

Reviewer 3 Report

The Authors have extensively revised the manuscript. The current version of the review is improved both in structure, clarity, and in the extent of the information provided.

This manuscript is a resubmission of an earlier submission. The following is a list of the peer review reports and author responses from that submission.

Round 1

Reviewer 1 Report

The review by Zhao and Saynal addresses the potential use of p53 isoforms as biomarkers or therapeutic targets for cancer. The topic is timely and interesting, and the references are up-to-date. However, the review needs major improvements to be of interest for a reader.

Major comments:

1- Overall, the review is too lengthy and the information not summarized enough. It is very difficult to understand the flow and message of the authors. The general feeling is that the authors pasted together pieces of information without linking them together to develop a narrative or an idea.

2- The English need major improvement and the text is full of grammar errors and poor choice of words. It would benefit from a revision by a senior author fluent in English.

3- There are multiple imprecisions or simplifications all over the review which make the message confusing.

For instance, line 15: the authors write “as the short isoforms (…) lacking the TADS binding to MDM2, they may not be degraded through ubiquitination”. MDM2 does not only bind to p53 TAD, but also to other parts of the p53 N-terminus and to p53 core domain (which is enough to increase p53 ubiquitination, see PMID16866370).

4- The presentation of the p53 downstream signaling pathway in part3 is very confusing and incomplete, and should be rewritten. The authors mention only p53 “canonical” functions (apoptosis and cell cycle control), which has been proven to not be enough to explain p53 tumor suppression. The authors should both summarize it more and be more general and cover more of the p53 pathway, instead of describing very little parts of it in details.

Minor comments:

Some typos (among others):

  • line 258 (Figure 3B): apoptisis instead of apoptosis
  • Line 270 (Figure 3B legend): NF-kB instead of NF-κB (kappa)

Reviewer 2 Report

The authors have done a very good job in summarising the literature on p53 isoforms. However, there are minor points that need to be addressed. These are as follows:

“The human TP53 gene is located on the human chromosome 17p13.1 and it 32 comprises of 13 exon…”. The TP53 genes has only 11 Exons of which 10 are coding. This is a long sentence with too many ideas, and confusing, this needs to fixed. Please correct. Also correct the number of exons in the Figure 1 legend. It should be 11 as is shown in the figure.

“D40p53 isoform lacks the TAD-1 but still contains the TAD-2 and it plays a positive or 58 negative role in cancer development”… Authors need to be specific.

“Recent research shows the role of RNA in aggregation of the core DBD of p53 isoforms and also FL53”, Once again which RNA? This sentence is confusing, please be specific.

“p53 isoforms were first identified in the mid-1980s in human and mouse during study 80 of p53 expression” This is incorrect. p53 isoforms were identified in the 2000s.

“They are extremely susceptible to decay (NMD) that can degrade premature 135 translation termination codons (PTCs)”.. please specify, nonsense mediated decay…

“Intriguingly, by interfering p53-MDM2 binding,…” please add the interfering with…

“The excess 236 of D40p53 compared to p53α exerts a bad effect on the cancer cells [24]”.. what is the bad effect, please specify…

Figure 3A , should demonstrate what p53a does? Also d133p53 inhibits p21 and miR-34 – but does not promote senescence. The arrows need to be corrected.

Section 3 title could be modified: Influence of p53 isoform network on biological processes

Also here would be a good place to refer to “Adaptive homeostasis and the p53 isoform network” as it covers quite a bit of the role of p53 network in detail.

It  would benefit from sub-sections.

3.1: p53, isoforms and apoptosis via 14-3-3

3.3: p53 isoforms, NFkb, JAK-STAT

“p53 isoforms can also mediate its network by interplaying with other factors. p53 314 isoforms can interact with p53 family proteins p63/p..: This paragraph could benefit from being moved up before the sub-sections.

“sequencing, which can- 347 not distinguish three variant transcripts α, β, and γ” : be specific: 5’ variants of the α, β, and γ transcripts – modify

However, the correlation between mRNAs and the abundances of their corresponding proteins is weak [127,128]. Is this a general observation or specific for p53 isoforms.. authors need to clarify..

“Additionally, in mucin- 360 ous ovarian cancer (MOC), Δ40p53 expression that is not detected in HGSOC raises the 361 recurrence-free survival rate [28,130].” Confusing sentence, talks about 2 different tumour types with unclear associations? Please modify.

“The increased expression of Δ133p53β predicted poorer outcomes in glioblastoma, mela- 371 noma, breast, and prostate cancer patients [125,126,132,133].” Yes it is true for glioblastoma and prostate? Not sure it is d133p53b for breast and melanoma. This needs to be corrected.

In addition, patients expressing 406 p53γ isoform in p53 mutant breast cancer have the good prognosis as those in wild-type 407 p53 breast cancer [144]. An additional paper that supports this hypothesis is “Regulation of the interferon-gamma (IFN-γ) pathway by p63 and Δ133p53 isoform in different breast cancer subtypes” This should be included.

Table 1 should be cited in section 4.

Reviewer 3 Report

The Authors aimed to provide an updated review on p53 isoforms, focusing on their potential role as cancer biomarkers and targets for therapeutic interventions.

In principle, the topic is timely and relevant to the journal readership. There are some good features in this review. For example, I found the figures to be generally clear and effective. There are also shortcomings that strongly reduce, in my opinion, the impact and value of the manuscript.

I’m referring mainly to redundancies, repetitions, typos, language oddities, and the use of vague terms or statements that limit the knowledge-gain of reading the work. I strongly invite the Authors to restructure and revise their manuscript profoundly.

Here is a list of places to start with as, I believe, they need particular attention:

Line 69, I’m not clear why the sentence starts as “in addition” …the concept summarized there seems opposite to the previous one

Line 82, something is missing in the sentence

Line 83, check the sentence

Line 94, why “a” DNA damage response?

Line 102 and 103 (and elsewhere in the text, e.g., 324), please specify what you mean by “regulate”, given that there are reports of both p53-dependent upregulation and repression of target genes

Line 135: consider specifying what you mean by “they”; since you are switching from a post-translational to a post-transcriptional regulation mechanism in the two connected sentences

Line 198: please clarify the sentence

Line 200: please clarify what you mean by “monitoring the genes”

Line 231: I’m not clear why you define the interaction as “cooperation”

Line 237: please explain what you mean by “bad effect”

Line 239-240: check the sentence

Line 267: check the sentence

Line 279: consider clarifying the underlying mechanism

Line 282-283: something is missing in the sentence

Line 293: “membranes”?

Lines 327, 328, 331: check sentences

Line 356: “high specific”

Line 358: can you say that an increase in D133 “improves” OS?

Line 384: I’m not clear about the use of “improves” in that context

Line 392: “These”?

Line 407: check the sentence

Line 412: check the sentence

The first paragraph of section 5 is a bit redundant with sentences in the Introduction

Line 427: “more novel ITAFs”?

Line 440: check sentence

Line 444: “molecule” or something is missing

Please check the column headers in Table 1 (for example, in column two, there is information on single isoforms, a combination of isoforms, and isoform ratio). Is “upregulated expression” clear? What do you mean by “detecting level”? Is it clear what “improve tumor development” means?
Line 451 can caffeine be called “targeted”?

Line 454, check the article

Line 460: check sentence

Line 484: is cancer “prevention” the goal?
